# The Complexity of Sesquiterpene Chemistry Dictates Its Pleiotropic Biologic Effects on Inflammation

**DOI:** 10.3390/molecules27082450

**Published:** 2022-04-11

**Authors:** Narcy Arizmendi, Syed Benazir Alam, Khalid Azyat, Darren Makeiff, A. Dean Befus, Marianna Kulka

**Affiliations:** 1Nanotechnology Research Centre, National Research Council Canada, Edmonton, AB T6G 2A3, Canada; narcy.arizmendipuga@nrc-cnrc.gc.ca (N.A.); sbalam@ualberta.ca (S.B.A.); khalid.azyat@nrc-cnrc.gc.ca (K.A.); darren.makeiff@nrc-cnrc.gc.ca (D.M.); 2Alberta Respiratory Centre, Department of Medicine, Faculty of Medicine, University of Alberta, Edmonton, AB T6G 2E1, Canada; dbefus@ualberta.ca; 3Department of Medical Microbiology and Immunology, Faculty of Medicine, University of Alberta, Edmonton, AB T6G 2E1, Canada

**Keywords:** sesquiterpenes, inflammation, dendritic cells, mast cells, immune responses

## Abstract

Sesquiterpenes (SQs) are volatile compounds made by plants, insects, and marine organisms. SQ have a large range of biological properties and are potent inhibitors and modulators of inflammation, targeting specific components of the nuclear factor-kappaB (NF-κB) signaling pathway and nitric oxide (NO) generation. Because SQs can be isolated from over 1600 genera and 2500 species grown worldwide, they are an attractive source of phytochemical therapeutics. The chemical structure and biosynthesis of SQs is complex, and the SQ scaffold represents extraordinary structural variety consisting of both acyclic and cyclic (mono, bi, tri, and tetracyclic) compounds. These structures can be decorated with a diverse range of functional groups and substituents, generating many stereospecific configurations. In this review, the effect of SQs on inflammation will be discussed in the context of their complex chemistry. Because inflammation is a multifactorial process, we focus on specific aspects of inflammation: the inhibition of NF-kB signaling, disruption of NO production and modulation of dendritic cells, mast cells, and monocytes. Although the molecular targets of SQs are varied, we discuss how these pathways may mediate the effects of SQs on inflammation.

## 1. Introduction

Globally, traditional medicine has employed empirical practices to gather knowledge and skills to treat different types of inflammation with several derivatives from plants. However, the inherent toxicity and the complex mixtures of such derivatives have prevented the proper exploitation of these natural resources. Careful scientific studies have identified specific sesquiterpene (SQ) molecules that could be used to treat different inflammatory-related diseases. Although there are previous reviews for the use of SQ to treat inflammatory responses, a clear definition of functional chemical groups or modes of action of SQ is unavailable. Our approach in this review is to focus on the structural characteristics, biogenesis, and biological activity of specific SQ and on inflammatory cells such as dendritic cells, monocytes, lymphocytes, and particularly mast cells, and allergic inflammation. We emphasize the effects of SQs on pathways that modulate inflammatory processes involving the nuclear factor-kappaB (NF-κB), nitric oxide (NO), and other more specific signaling pathways. We offer the reader the latest information about specific molecules, with stereospecific configurations and their molecular targets that could be relevant in the modulation of inflammation.

SQs and SQ lactones are a large class of biologically important terpene derivatives produced by plants [1]. SQs are composed of a 15-carbon backbone that forms the basis of a large diversity of cyclic and acyclic structures. SQs can be isolated from several plant families including *Asteraceae* and *Compositae*, from over 1600 genera and 2500 species, by using a variety of approaches including silica gel filtration and chromatography [2]. Every year, several new SQ are isolated and tested for their potential therapeutic applications. To date, over 500 different SQs have been identified, including more than 100 known derivatives reported from plants. Additionally, several strategies have been developed to synthesize SQs with different functionalities which exploit their many chemical structures.

Although some of these SQs have been used clinically to treat patients in some countries, we have only begun to understand the molecular mechanisms of action of most SQs. Several recent publications have described the anti-inflammatory properties of SQs, emphasizing their effect on lymphocyte differentiation and regulation of cytokine production. However, there is still much to learn about the activity of SQs as immunomodulators because there is a mixture of studies using naturally derived compounds, either highly enriched or only crude extracts, and synthetic compounds of varying purity. Moreover, studies that have examined SQs as biomedical targets have used a diverse range of biological readouts in several different cell types but have provided few focused studies on specific biochemical pathways. Specific molecular targets are difficult to determine because the analyses are often focused on in vivo animal studies or cell function studies with poorly defined endpoints. Often, the language used to define SQ effects are vaguely described as “anti-inflammatory” without clearly indicating which cells, signaling pathways, or tissues are involved. It should be noted that inflammation is not a single process but a series of events that are initiated under diverse conditions and lead to a plethora of clinical manifestations. Even when SQ compounds seem to decrease some aspects of inflammatory responses, their poor solubility, dosage, and stability limit widespread clinical application.

Several recent reviews have examined the effect of specific types of SQs on inflammation, and have examined the structure/function relationship of several specific types of SQs such as SQ lactones [3,4,5,6]. In this review, we focus mainly on the cells and pathways involved in allergic inflammation. We will examine some of the sources, chemical properties and anti-inflammatory effects of specific SQs with the objective to better understand this complex group and their potential application as anti-inflammatory compounds mediated by allergic pathways. Our focus will include biological effects on cells directly involved in allergic inflammation such as mast cells, monocytes, and dendritic cells and will focus on inflammation initiated and facilitated by the key transcription factor nuclear factor kappa B (NF-kB) and nitric oxide (NO). Whenever therapeutic applications are mentioned, it is only in the context of possible biological outcomes, and a comprehensive analysis of clinical studies is beyond the scope of this review. Any conclusions on the clinical use of these compounds are for illustrative purposes when discussed in the context of biochemical pathways.

## 2. SQs Are Plant Metabolites That Target Several Molecular Signaling Pathways

### SQs Are a Plant Defense Molecule Produced in Response to Stress

SQs are an ancient class of compounds produced by plants, insects, and marine organisms. In plants, SQs comprise the complex protective system of metabolites that are produced in response to both biotic and abiotic stress (Figure 1). SQs protect plants from stressful environmental conditions and prevent pathogen invasion [1]. SQs are produced in the leaves and flowering heads of plants [1] from several orders including: magnoliales, rutales, cornales, and asterales. Several SQs isolated from tomato, tobacco, and potatoes have also been classified as phytoalexins [7,8], which are de novo synthesized antimicrobial compounds produced by plants as a response to biotic or abiotic stress [9,10,11]. SQs are constitutively produced in the plant cytosol [12] and in some species are stored in trichomes, thin epidermal outgrowths that often appear as fine hairs or spikes [1]. SQs are generally present in low levels in healthy plants [13], but upon infection by certain pathogens, the biosynthesis of SQs increases rapidly and can be sustained over several days [14,15].

The biosynthesis of SQs can be induced by bacterial and fungal infection [8], and SQs can target and neutralize such pathogens. SQ compounds comprise a significant proportion of a plant’s biogenic volatile organic compound response, which plays a prominent role in plant defense as well as inter- or intraplant signaling (Figure 1) [1,16,17]. SQs not only can disrupt tobacco mosaic virus [16,17,18] infection of plants but can also inhibit in vitro replication of several animal viruses including influenza virus [19,20], human cytomegalovirus, measles virus, and herpes simplex virus type-1 [21]. Not only do SQs protect plants from insects and parasites mainly by acting as a feeding deterrent due to their extremely bitter taste [22,23], but SQs can also interfere with an insect’s metabolism and disrupt communications in their central nervous system [24,25]. The SQ α-farnesene, which is responsible for the characteristic smell of apples (*Rosaceae*), has a dual function as chemoattractant and chemorepellent, in attracting feeders such as birds to facilitate the spread of seeds, and acting as a potent insecticide, respectively (Figure 1) [1]. Hypotheses have been proposed involving the release of volatile SQs by plants to attract predators that feed on parasites. Thus, predators are cleverly used to clear the parasite infestations [26], a phenomenon known as indirect plant defense or “cry for help” [27,28,29].

SQs are also produced and released by plants as a result of abiotic stress [30,31] such as drought [32], heat [33], and ozone damage [34]. One of the consequences of abiotic stress is the accumulation of reactive oxygen species (ROS) which can be detrimental to the growth of the plant [35], but SQs can quench ROS by reacting with the unsaturated SQ hydrocarbon skeleton and thereby can serve as a potent antioxidant in plant protection [36] (Figure 1). This ROS scavenging property of SQ is also attributed to the high reactivity of SQs with ozone [37].

SQs are able to target protozoal, fungal, bacterial, and viral microorganisms, demonstrating their versatility and ability to target several molecular pathways. This diversity is dependent upon their structural diversity. In the next section, we will examine the main structural constituents of most SQs and a few examples of biosynthetic pathways that lead to this complexity.

## 3. Chemical Structure and Biosynthesis of SQ and SQ Lactones

SQs can be divided into SQ and SQ lactones, both of which have several defining features.

### 3.1. Structures of SQs

The basic chemical structure of SQs consist of the terpene subunit (C_5_H_8_)*_n_* where *n* = 3 (i.e., C_15_H_24_). The SQ scaffold represents extraordinary structural variety consisting of both acyclic and cyclic (mono, bi, tri, and tetracyclic) compounds, which can be decorated with a diverse range of functional groups and substituents in distinct regions and stereospecific manners. SQs are naturally present as hydrocarbons or oxygenated molecules such as alcohols, ketones, aldehydes, or lactones. Some SQs contain several chiral centers and various substituents which result in a complex chemical structure for a C_15_ compound. Over 300 different carbon skeletons and about 10,000 SQs have been described and all are derived from the acyclic SQ farnesane [38]. All of these derivatives are formed via cyclization reactions mediated by SQ synthases in biosynthetic cascades initiated by the formation and propagation of highly reactive carbocation intermediates (Figure 2) [39].

### 3.2. Structures of SQ Lactones

SQ lactones are cyclic SQs with a fused α-methylene-γ-lactone ring. Subclasses include eudesmanolide, guaianolide, pseudoquaianolide, germacranolide, and xanthanolide (Figure 3) [41]. SQ lactones are formed via a mevalonic acid pathway by plants [42] (Figure 4). The first step of the mevalonate pathway is the condensation of two acetyl-CoA molecules catalyzed by an acetoacetyl-CoA enzyme. The second step results in the formation of 3-hydroxy-3-methylglutaryl-CoA (HMG-CoA) by the intervention of HMG-CoA synthase (HMGS). The third step consists of the conversion of HMG-CoA to mevalonate (MVA) by the enzyme HMG-CoA reductase (HMGR). The final product, isopentyl phosphate (IPP), is obtained after three successive steps catalyzed by the mevalonate kinase (MVK), phosphomevalonate kinase (PMK), and mevalonate diphosphate decarboxylate (MDD) and then converted to dimethyallyl diphosphate (DMAPP). Finally, IPP and DMAPP are converted by farnesyl diphosphate synthase to farnesyl diphosphate (FPP) which is the common precursor for the formation of SQ lactones (Figure 4) [43,44,45]. The most common feature of SQ lactones is the presence of a γ-lactone ring bearing an α-methyl group. SQ lactones are generally present in very low quantities in plants, but the percentage of SQ lactones can represent up to 5% of the dry weight of the plant depending on the plant species [46].

SQ lactones with pharmacological properties that have been isolated from plants are generally colorless compounds and, although they exhibit poor pharmacokinetic properties, they have been used successfully as drug therapies. One such example is arteminsinin, shown in Figure 3 and discussed in more detail later in this review, and its semi-synthetic SQ artesunate, a derivative isolated from *Artemisia annua*.

## 4. SQ as Biologically Active Molecules—Role in Inflammation

The literature is replete with descriptions of the biological activities SQs. Over the past few decades, several biologically active SQ compounds have been isolated from *Centaurea* species (*Asteraceae*) and purported to have neuroprotective, antioxidant, and antimicrobial effects in vitro [47]. Costunolide, isolated from *Centaurea*, *Campomanesia* (*Myrtaceae*), and *Petasites* (*Asteraceae*), has some anti-allergic [48], anti-microbial [47,49], anti-diabetic [50], and neuroprotective bioactivities [51]. SQs from the *Asteracea* family species *Cichorium intybus*, *Centripeda minima*, and *Petasite hybridus* have been used to treat several ailments, such as liver disease [52,53], gastrointestinal disorders [54], diabetes, and malaria [55], and they exhibit antimicrobial, anti-parasitic, hepatoprotective, anti-diabetic, anti-tumoral, analgesic, antioxidant [56,57,58], and anti-inflammatory effects (Table 1). However, these studies have often been vague, using cellular assays which do not adequately examine the molecular or genetic targets of SQs. As a result, it is difficult to adequately summarize all of these effects in this review with any kind of comprehensive analysis of the mechanisms of action of SQs. For this reason, we have decided to focus on the effects of SQs on inflammation. Because inflammation is itself a complex phenomenon that has varied etiologies and tissue-specific processes, we will focus on inflammation mediated by the transcription factor NF-kB, the immunomodulatory gas NO, and tissue-resident immune cells such as dendritic cells and mast cells.

### 4.1. Isolation and Purification of SQ

The isolation and purification of SQsSQ from plant extracts is difficult, and complex extraction methods generate low yields and high cost [81]. Thus, many research groups synthetize the organic compounds and may work to purify these compounds for study [82,83]. Many of these SQ target different cellular pathways. For example, parthenolides disrupt tubulin polymerization and thereby inhibit cell division and proliferation [84]. Because these pathways are too numerous to discuss in this review, we will focus on some of the most prominent pathways that are associated with inflammatory diseases.

It is possible that SQs function in synergy with many of these other compounds in extracts to exert their biological effects; thus, purification and isolation of specific SQs may be counterproductive in terms of therapeutic applications. However, only studies of purified compounds alone, and then in combinations with others, will define their mechanism of disrupting inflammatory pathways, and thus we will attempt to focus the majority of this review on studies that have used synthetic or purified SQs, whenever possible. Table 1 shows the most representative anti-inflammatory effects of some SQs wherein the bioactive SQs are wholly or partially identified. For the remainder of this review, we will refer to these compounds and their biological actions on the two main inflammatory pathways mediated by NF-kB, NO and specific inflammatory cells.

### 4.2. Mechanisms of Action

Biological applications of purified SQs face two main challenges. First, purified SQs are often insoluble in water and therefore difficult to test in in vitro systems and difficult to administer to in vivo models of disease. Second, SQs have poor bioavailability at low concentrations and are cytotoxic at high concentrations [85]. The low bioavailability is due to high body clearance [86]. Recently, Garcia et al. [87] found that the SQ lactones lactucopicrin and lactucin from curly escarole (Asteraceae) are orally bioavailable, but undergo gut microbiotic and phase II metabolism in humans, as determined by qualitative analysis of SQ lactones and metabolites in urine. Another later study reported that the poor bioavailability of brussels/witloof chicory SQ lactones in humans was due to gut microbial and phase II metabolism [88]. Therefore, much of the research describing the effects of SQs in various biological assays must be carefully considered with respect to the concentrations and purity administered. Additionally, it is worth noting that responses to natural products are often biphasic, meaning that lower concentrations can sometimes have more profound effects on the signaling pathways involved in a particular biological function. Regardless, the current literature suggests that many SQs function by blocking some common inflammatory pathways, particularly those involving two transcription factors, the nuclear factor of the κ chain in B-cells (NF-κB), and nuclear factor of activated T-cells (NFAT), which are required for the gene transcription and expression of a variety of pro-inflammatory mediators such as tumor necrosis factor (TNF).

#### 4.2.1. NF-κB and NFAT Signaling in Inflammation and Its Modulation by SQs

The nuclear factor of the κ chain in B-cells (NF-κB) is expressed in nearly all cells and regulates the expression of genes related to cellular interactions, survival, differentiation, and adhesion, and specifically initiates the expression of cytokines, chemokines, and coagulation factors. One of its most prominent roles is the initiation and potentiation of inflammation (Figure 5). The NF-κB transcription factor family consists of five different DNA-binding proteins that form several homodimers and heterodimers [89] and activation of NF-κB depends on the degradation of specific inhibitors, such as the inhibitor of NF-κB (IκB), following phosphorylation by the IκB kinase (IKK) complex. Increased NF-κB activity in inflammatory reactions is due to augmented production of IKK-activating cytokines, including tumor necrosis factor (TNF), and interleukin 1 (IL-1) [90]. The genetic responses following NF-κB activation depend on the cell type, the accessibility of promoter regions that are regulated by epigenetic mechanisms, and the dynamic signaling networks which involve significant crosstalk between several upstream and downstream pathways that may have additional implications in inflammation [91].

Several SQ compounds modulate the inflammatory response by inhibiting the NF-κB pathway [61] (Table 1), leading to suppression of cyclooxygenase-2 (COX-2) and increased expression of anti-adhesion mucin MUC-1 [92]. In general, SQ inhibit the phosphorylation of IκB, preventing activation and further translocation of NFκB subunit p65 (RelA), preventing binding to DNA and consequent transcription of inflammatory cytokines such as TNF and IL-6 (Figure 3). SQ lactones also induce c-Jun NH_2_-terminal kinase (JNK) activation independent of NF-κB inhibition [93], and can modify the activity of p53 by two mechanisms: (1) by promoting the ubiquitination and degradation of the mouse double minute 2 homolog (MDM2), a p53 negative regulator, or (2) by depletion of histone deacetylase 1 (HDAC1) through proteasomal degradation [94,95]. In addition, artesunate, an artemisin-derived SQ, inhibits the signal transducer and activator of transcription (STAT) proteins, by blocking STAT3 phosphorylation [96]. Experimental studies have shown an improvement in the inflammatory responses after the use of SQ, and this improvement has been associated with NF-κB inhibition. However, studies on specific molecular targets of SQ have not been completed.

It has been shown that dehydrocostuslactone, costunolide, and alantolactone, counteracted the pro-inflammatory effects of TNF-stimulation, IFN-γ-stimulation, and IL-22 on keratinocytes via inhibition of STAT1 signaling [97], and suppressed inducible COX-2 expression by downregulating NF-κB, MAPK, and AP-1, which are critically involved in the signal transduction of pro-inflammatory cytokines in macrophages [98], colorectal cancer cells and keratinocytes [99]; 1β-hydroxyalantolactone (IJ-5) SQ lactone from *Inula japonica* (*Asteraceae*) suppressed TNF-induced NF-κB activation and inflammatory gene transcription by inhibiting the ubiquitination of receptor-interacting protein 1 and NF-κB essential modifier, preventing IκB kinase activation [78]. Micheliolide inhibited intestinal inflammation mediated by NF-κB activation and subsequent pro-inflammatory pathways activation in vitro [100]. SQ lactones from *Neurolaena lobate* (*Asteraceae*) inhibited lipopolysaccharide (LPS) and TNF production, interfered with the production of E-selectin and interleukin-8 (IL-8) in HUVECtert and THP-1 cells, and inhibited the development of acute inflammation in carrageenan-induced paw edema in rats [59].

In addition to NF-kB, the nuclear factor of activated T-cells (NFAT) family of transcription factors is also targeted by SQs. Once NFAT is dephosphorylated, it translocates to the nucleus and controls the expression of several genes. NFAT also modulates diverse cell functions, and dysregulation of NFAT contributes to the development or prevalence of chronic inflammatory and autoimmune diseases. FK506 or tacrolimus interacts with FK506-binding protein-12, inhibiting the dephosphorylation of NFAT by calcineurin, a serine/threonine phosphatase. Several natural extracts from *Arnica montana* (*Asteraceae*) containing SQ lactones from the pseudoguaianolide family inhibit NFAT-DNA binding [101].

#### 4.2.2. Production of NO in Inflammation, Inflammatory Markers, and Its Modulation by SQ, and Subsequent Downstream Effects

NO is a ubiquitous physiological mediator synthesized by many cell types, functions as a secondary messenger, and acts as an important regulator of inflammation. NO activates and regulates the function of immune cells, sometimes inducing apoptosis [67]. Since NO is lipophilic, highly diffusible, and short-lived, it is an ideal messenger capable of regulating intercellular and extracellular signaling pathways. Pleiotropic effects of NO include the modulation of vasodilation, respiration, neurotransmission, cell migration, immune responses, apoptosis, and metabolism. At a subcellular level, NO modifies mitochondrial respiration, increases glutamine consumption in the tricarboxylic acid cycle, and in some instances increases tumor growth and confers chemoresistance [102,103]. NO is also an anti-inflammatory or immunosuppressive molecule and functions by inducing the apoptosis of activated inflammatory cells, thereby turning off the inflammatory signal [104]. NO is synthesized by nitric oxide synthase (NOS) which has three isoforms: neuronal NOS (nNOS or NOS1), inducible NOS (iNOS or NOS2), and endothelial NOS (eNOS or NOS3). The principal enzyme involved in the NO pathway in activated immune cells is the inducible type-2 isoform of NOS-2, producing sustained NO synthesis from L-arginine [105]. NOS2 induction is independent of calcium concentration but can be stimulated by cytokines produced by all cell types. NOS2 often requires two signals for activation: one from interferon gamma (IFNγ), and another from tumor necrosis factor (TNF) [106]. In tumors, the hypoxia inducible factor-1α (HIF-α) interacts with IFNγ and induces NOS2 expression. The activation of NOS2 by TNF occurs via stimulation of NF-κB which binds to a κB element in the NOS promoter [107]. Recent clinical studies have shown a link between high levels of NOS2, the onset of progressive inflammatory disorders, and an elevated risk of developing cancer [108,109]. Several factors may generate high levels of NO such as its co-expression with COX-2 which catalyzes the conversion of arachidonic acid to prostaglandin E2 (PGE2). NOS2/COX2 crosstalk also involves NO-inducer biomarkers such as TNF, IL-6, and IL-8 which are implicated in cell signaling upregulating MAPK, PI3K, and STAT3 [110]. The role of NO in inflammation and neurobiology depends specifically on its concentration, its chemical reactivity, the vicinity of target cells, and the way that target cells are programmed to respond to different stimuli. In addition, NO exhibits pro- and anti-nociceptive effects in the nociceptive processing pathway, in response to potentially toxic stimuli depending on factors such as the site of action and NO concentration [111].

One of the most potent inhibitors of NO production is β–elemonic acid (β-EA) in both in vitro and in vivo models. Within the concentration range of 0.02–0.5 μM, β-EA significantly reduced NO production by LPS (1 μg/mL) treated RAW 264.7 cells and downregulated the production of several other mediators, including tissue inhibitor of metalloprotease 1 (TIMP1), TNF, IL-6, monocyte chemotactic protein 1 (MCP-1), soluble TNF receptor 1 (sTNFR1), eotaxin-2, interleukin 10 (IL-10), granulocyte colony-stimulating factor (GCSF), and macrophage inflammatory protein 1 alpha/gamma (MIP-1α/γ) [112]. In a carrageenan-induced paw edema in vivo model, β-EA reduced inflammation-associated edema by 40% as early as 30 min post-treatment, suggesting a very potent effect. The mice received 100–300 mg/kg of β-EA orally for 7 days, a very high dose for an in vivo treatment model, followed by the subcutaneous injection of 50 μL of 0.1% (*w*/*v*) carrageenan into the sub-plantar region of the right hind paw, and the β-EA anti-inflammatory effect was measured as a percentage of edema in a dose- and time-dependent manner. The carrageenan-induced edema generated an increase in PGE2 levels, and β-EA significantly decreased PGE2 levels in a dose-dependent manner [112] (Table 1). Following these studies, a pharmacokinetic, and tissue distribution analysis measured β-EA in plasma and tissues in a rat model after oral administration of 85 mg/kg β-EA [113]. After 1 h, β-EA was present in plasma, heart, liver, spleen, lung, kidney, jejunum, colon, and ileum, following a concentration trend: jejunum > ileum > colon > liver > kidney > lung > heart > spleen. The peak concentration was found in 4 h for most tissues, except in jejunum and colon wherein the concentration levels peaked at 1 h and 8 h respectively [113].

A SQ lactone from *Inula japonica* (*Asteraceae*) called JEUD-38 attenuated NO production of LPS-treated RAW 264.7 cells by blocking protein expression of NOS2 [114]. This decrease in NOS2 expression was the result ofa reduction in the translocation of NF-κB subunit p65, via IκBα phosphorylation and degradation. JEUD-38 also inhibited the MAPK pathway [114].

SQs derived from several plants used in traditional medicine, such as *Alpinia oxyphylla* (*Zingiberaceae*) have potent inhibitory effects on NO production in LPS-activated mouse peritoneal macrophages, and β–hexosaminidase release from a basophilic cell line, RBL-2H3 [65]. The Curcumol SQ from *Curcuma kwangsiensis* (*Zingibetaceae*) inhibited the VEGF/AKT/eNOS signaling pathway, and reduced expression of NO, inhibiting liver angiogenesis and mediating anti-fibrotic effects in liver [66]. SQ lactones from *Artemisia leucodes* (*Asteraceae*) showed hepato-protective properties, stimulating changes in NO metabolism in the monooxygenase and NOergic systems in hepatocytes in animals with acute toxic hepatitis [115]. In addition, alantolactone, a SQ lactone from *Inula helenium* (*Asteraceae*), decreased the expression of COX-2, iNOS, MMPs, and a disintegrin and metalloproteinase with thrombospondin motif 5 induced by IL-1β in mouse chondrocytes in an osteoarthritis model [116].

LPS-induced inflammation was also downregulated in mouse microglia by Atractylenolide-III from *Atractylodes macrocephalia* (*Asteraceae*). Atractylenolide-III suppressed the LPS-stimulation of MG6 cells, decreasing mRNA expression and protein levels of TNF, IL-1β, IL-6, iNOS, and COX-2 and inhibited the phosphorylation of p38 MAPK, and c-Jun NH_2_ terminal kinase (JNK) [117]. These effects may have some similarities to the ones observed in microglia, where several components isolated from *Pogostemon cablin* (*Lamiaceae*) patchoulol-type, and guaiane-type SQ inhibited NO production in LPS-stimulated BV2 microglia cells, suggesting a significant effect on neuroinflammation [118].

The possibility that SQs can alter NO in the brain and thus modulate neuroinflammation is attractive because many neurodegenerative diseases are thought to be caused by chronic smoldering neuroinflammation [119]. This is especially true for SQs that could potentially cross the blood brain barrier, which is a major hurdle for most current clinical therapies. The bicyclic sesquicentennial β-caryophyllene (BCP) has a high affinity for cell membranes and can alleviate ischemic brain damage in rats, reducing oxidative damage and neuronal apoptosis [51,120]. Thus, BCP can attenuate cerebral ischemia-reperfusion injury (CIRI) via regulation of numerous molecular targets by altering gene expression, signal transduction pathways, or direct interactions [121,122]. Although the precise molecular targets of particular SQs in CIRI remain unknown, it is entirely possible that NO production plays an important role. In this regard, recent studies by Liu et al., (2021) using transcriptome sequencing analysis, have begun to identify specific molecular mechanisms involved in the effects of SQs on CIRI in the rat. RNA-Seq technology allowed the construction of a protein interactive network, Hub genes, and a transcription factor (TF) regulatory network for the potential Hub genes [51]. Experimental studies identified 411 differentially expressed genes (DEGs) filtered by treatment (CIRI vs. Sham, and CIRI vs. BCP), allowing a determination of a cluster profile, with levels of Paired box 1, Cxcl3, and Ccl20 remarkably inhibited by BCP treatment. Gene ontology and Kyoto encyclopedia of genes and genomes (KEGG) pathway analysis, identified significant DEGs involved in multiple biological processes such as extra-cellular matrix organization, leukocyte migration, regulation of angiogenesis, and reactive oxygen species metabolic processes [51]. DEGs also participated in several signaling pathways including MAPK signaling pathways, cytokine–cytokine receptor interaction, JAK-STAT signaling pathways, and others. The protein–protein interaction network consisted of 339 nodes and 1945 connections, and the top ten Hub genes identified included TIMP1, MMP-9, and STAT3. These studies also established a TFs-miRNAs-targets regulatory network involving 6 TFs, 5 miRNAs, and 10 Hub genes, consisting of several regulated signaling pathway models such as Brd4-rno-let-7e-Mmp9, Brd4-rno-let-7i-Stat3, and Hnf4a-rno-let-7b-Timp1. Western blot analysis demonstrated that BCP inhibited TIMP1, MMP-9, and STAT3 expression in rat brains after ischemia/reperfusion. ELISA analysis showed that BCP suppressed TNF and IL-1β production during CIRI, and suppressed oxidative damage. Altogether, BCP significantly reduced neurological deficit and improved cerebral ischemia [51].

## 5. SQ Effects on Inflammatory Cells and Their Activation in Allergic Inflammation

Although NF-kB, NFAT and NO are general regulators of inflammation, these pathways can be initiated in different ways depending on the stimulus and the tissue in which it is initiated. Allergic inflammation is a complex interplay of proinflammatory cells and mediators that reside mainly in tissues that interface with the external environment, such as the mucosal surfaces of the lung, gut, and the skin. This section discusses the effects of SQs on three important types of cells (dendritic cells, monocytes, and mast cells) in allergic inflammation.

### 5.1. SQ Effects on Dendritic Cells, Monocytes and Lymphocytes

Dendritic cells play a major role in inflammatory processes, in detecting and binding infectious agents, migrating to secondary lymphoid tissues and priming naïve T lymphocytes, leading to the development of diverse immune responses (Figure 5). Conversely, they can also prevent immune responses, as they may be involved in central and peripheral tolerance. “Inflammatory dendritic cells” have recently been redefined as dendritic cells that are differentiated from monocyte cells and play an important role during infection or inflammatory processes in innate and adaptive immunity [123]. Several research groups have focused on dendritic cells derived from the monocyte lineage of mice and humans and have identified two functional subsets defined by the expression levels of CX_2_CR1. As described by Geissmann and colleagues [124], resident CX_3_CR1^high^ monocytes are found in the blood and inflamed peripheral organs. On the other hand, CX_3_CR1^low^ monocytes are short-lived and are actively recruited into inflamed tissues, where they further differentiate into functional DCs that have the ability to stimulate naïve T cells. Studies from our group have shown that novel isolated SQ molecules [125], fukinone, and 10βH-8α,12-epidioxyeremophil7(11)-en8β-ol, derived from *Petasites tatewakianus* (Asteraceae), downregulate dendritic cell functions mediated by proliferator activator receptor gamma (PPARγ), hindering the expression of CD86 and the production of TNF, IL-6 and IL-12p70 after LPS stimulation, inhibiting maturation and activation of bone marrow-derived dendritic cells [80]. Conversely, the SQ called myrothecine A found in *Artemisia annua* (Asteraceae) has effects on the proliferation of the mouse hepatocarcinoma cell line Hepal-6, and the maturation of dendritic cells by regulating the oncogene miR-221, and enhancing the expression of CD86 and CD40. Myrothecine A acts as an anti-tumor drug by promoting maturation of dendritic cells in the environment of hepatocellular carcinoma [126]. SQs can suppress autophagy in hepatocytes and Kupffer cells isolated from an obese mouse model in which there is an increase in TNF production, and loss of autophagy enhanced endotoxin-induced inflammasome activation, leading to the production of IL-1β, and IL-18 [127,128]. Moreover, the SQ called micheliolide (MCL) alleviated hepatic steatosis in the *db/db* mouse model by inhibiting inflammation and lipid accumulation in lipid mixture-induced AML12 and L02 cells by upregulating PPARγ and decreasing phosphorylation of IκBα and NF-κB/p65, inhibiting NF-κB (Table 1). In addition, MCL administration increased microtubule-associated proteins 1A/1B light chain 3B, autophagy related 7 and Beclin-1 expression and the autophagosomal marker 3B-II/I ratio in *db/db* mouse livers and lipid mixture-treated AML12 and L02 cells. MCL-induced increases were mediated by the activation of PPARγ, AMP-activated protein kinase, and inhibition of phospho-mammalian target of rapamycin signaling [68].

Verticillane-type diterpenes, and some eudesmane-type sesquiterpenoids derived from *Cespitularia* sp. (*Xeniidae*) suppressed the release of TNF and NO and inhibited the upregulation of pro-inflammatory iNOS and COX-2 genes in LPS-induced dendritic cells [129]. Recent studies have shown that custunolide, the main active constituent of *Radix aucklandiae* (*Asteraceae*), possesses anti-inflammatory and immunomodulatory effects, downregulating myeloperoxidase activity, decreasing the pathological changes, and the levels of proinflammatory cytokines in mouse models of colitis. Custunolide also rebalanced Th17/Treg cells in the colon, mesenteric lymph nodes and spleen, while also reducing mRNA expression of *Rorc*, and *IL-17a*. Although there were no effects on dendritic cell maturation in vitro, mRNA expression of *Ifng and IL-6*, and Treg cell differentiation, there was a significant decrease in Th17 cell differentiation upon SQ treatment [130].

### 5.2. SQ Effects on Mast Cells/Basophils and Allergic Inflammation

Mast cells and basophils are hematopoietically derived cells that participate in inflammation, tissue remodeling, and infection, and are potent effector cells in innate and adaptive immune responses (Figure 6). Reaching sizes up to 20 μm, with a round nuclear morphology, mast cells contain abundant metachromatic cytoplasmic granules, due to their sulfated proteoglycans (i.e., heparin and chondroitin sulfates) [131]. Two subtypes of human mast cells have been described based on whether they contain specific proteases in their granules. Mast cells that store only tryptase (MC_T_ cells) are mainly found in the mucosa of the respiratory and gastrointestinal tracts and their numbers increase with mucosal inflammation. On the other hand, mast cells that contain tryptase and mast cell-specific chymase (MC_TC_ cells) are localized within connective tissues such as the dermis, submucosa of the gastrointestinal tract, heart, conjunctivae, and perivascular tissues [132]. Both of these types of mast cells are associated with blood vessels and epithelial surfaces and are central to the pathogenesis of immediate hypersensitivity and mastocytosis. In addition, mast cells express immunoglobulin Fc receptors (FcεRI, FcγRI, FcγRIIα, FcγRIIβ, and FcγRIIIα), complement receptors (C3aR, C5aR) and cytokine receptors (KIT, IL-3R, IL-18R, IL-33R, and TSLPR). When activated, mast cells produce IL-3, IL-4, IL-5, IL-6, IL-8, IL-9, IL-10, IL-13, IL-15, IL-25, TSLP, TNF, TGFβ, NGF, βFGF, VEGF, and CCL2 chemokines, and are involved in host responses to pathogens, autoimmune diseases, fibrosis, and wound healing [133]. Finally, mast cells typically have long lifetimes, remaining in tissues for months and may have the ability to proliferate and self-renew throughout the lifetime of the individual [134].

Basophils are smaller than mast cells at 5–8 μm in diameter, and contain a segmented, condensed nucleus, with fewer but larger granules compared to mast cells. After exiting the bone marrow and entering the circulation, basophils do not proliferate and have a short half-life of 2 to 3 days under steady-state conditions, but can be recruited to inflammatory sites in response to allergic stimuli [135]. Basophils express a variety of cytokine receptors (i.e., IL-3R, IL-5R, IL-18R, IL-33R, TSLPR, and granulocyte-macrophage colony-stimulating factor receptor), chemokines (CCL3, CCL4, CCL12, CXCL2), complement receptors (C3aR, and C5aR), chemoattractant receptor-homologous molecule on T helper type 2 cells, and Fc receptors FcεRI and FcγRIIIα. Both mast cells and basophils share some biomarkers, including the high-affinity IgE receptor FcεRI, but these cells develop from distinct hematopoietic progenitors and follow different differentiation pathways. Although a great deal of work has shown the effects of SQs on mast cells, very little is known of the effects of SQs on basophils.

*Petasites japonicus* (*Asteraceae*) is used to treat asthma and allergic diseases in traditional Korean, Japanese, and Chinese medicine. One SQ isolated from *Petasites* (*Asteraceae*), petatewalide B, inhibited antigen-induced degranulation of RBL-2H3, with no effect on Ca^2+^ levels [136]. In mouse peritoneal macrophages petatewalide inhibited LPS-induced iNOS, and nitric oxide production, but not COX-2 [137]. In an ovalbumin-induced asthma mouse model, petatewalide B reduced the number of eosinophils, macrophages, and lymphocytes in the bronchoalveolar lavage fluid following ovalbumin challenge, suggesting that petatewalide B may have reduced the production of mast cell-derived chemokines responsible for recruitment of these cells to the lung [136]. Because the transcription of many of these chemokines is dependent upon NF-kB and NFAT, it is possible that SQ blockade of these transcription factors may be responsible.

SQs also inhibit basophil degranulation, which is a process that is independent of transcription. SQ lactones isolated from the roots of *Saussurea costus* (*Asteraceae*) inhibited antigen-induced degranulation of RBL-2H3 cells (Table 2). Moreover, when applied in an ovalbumin-induced allergic asthma model, these SQ lactones decreased the number of immune cells such as eosinophils in bronchoalveolar lavage fluid [138], and reduced expression and secretion of the Th2 cytokines IL-4 and IL-13 in bronchoalveolar lavage fluid and lung tissues [139]. Our own studies demonstrated that RBL-2H3 cells sensitized with anti-DNP-IgE and activated by anti-IgE, the SQ lactone 6β-angeloyloxy3β,8-dihydroxyeremophil-7(11)-en-12,8β-olide (F-1) had no effect on degranulation when the cells were treated for only 30 min [140]. However, treatment of RBL-2H3 cell for 24 h with F-1 SQ lactones resulted in a dose-dependent suppression of degranulation and TNF production by as much as 90% (Table 2). These studies involved two synthetic versions of F-1, only differing in the hydroxylation of the third carbon, which suggested that the biological activity is dependent upon the particular chemical structure of F-1. Our study is one of the few that directly links SQ structure to immune cell function.

Another SQ compound, tussilagone isolated from the flower bud of *Tussilago farfara* (*Compositae*) [162] was tested in an ovalbumin-induced allergic asthma model in guinea pigs, in which 25 and 50 mg/kg of tussilagone was intraperitoneally administrated, and serum and nasal tissue samples were evaluated for features of inflammation [163]. A decrease in inflammatory cell infiltrates, lower histamine, ovalbumin-specific IgE, as well as IL-6 levels were observed after tussilagone administration. Conversely, TNF levels were increased after tussilagone treatment. IgE-stimulated RBL-2H3 cells treated with tussilagone in vitro showed suppressed phosphorylation of tyrosine kinase (Lyn), spleen-associated tyrosine kinase (Syk), Akt, NF-κB p65, extracellular signal-regulated kinase (ERK), and p38 MAPK [150] suggesting that tussilagone blocked FcεRI signaling, resulting in a reduction in transcription of pro-inflammatory mediators. Thethe in vivo effects of tussilagone in a murine model of asthma may be due to a decrease in ovalbumin-specific IgE, and inhibition of the IgE-producing B cells.

Artesunate, a semi-synthetic SQ derivative of artemisinin isolated from *Artemisia annua* (*Asteraceae*) has been used as an antimalarial and antitumor treatment, and inhibits IL-6, IL-17 and TNF secretion by synovial cells [164]. Bai et al., [96] showed that in an animal model of atopic dermatitis, artesunate attenuated 2, 4-dinitrochlorobenzene-induced atopic dermatitis in BALB/c mice, improving atopic dermatitis symptoms, and mast cell infiltration, as well as decreasing TNF, and IgE levels. The observed effects were related to downregulation of the Th17 cell responses, including modulation of IL-6, IL-17, IL-23, STAT3 and ROR-γt upon artesunate treatment.

In addition, the oral administration of tomentosin (*Inula* extract, family *Asteraceae*) reduced mast cell-mediated passive cutaneous anaphylaxis in IgE-sensitized mice. Tomentosin also reduced production of eicosanoids (PDG_2_ and LTC_4_) by stem cell factor (SCF)-stimulated bone marrow mast cells and inhibited mast cell degranulation by suppressing MAPK, PLA2, and PLCγ1 phosphorylation and intracellular calcium release [165].

## 6. Other Important Targets of SQs That May Modulate Inflammation

### 6.1. Effects of SQs on Ion Channels That Regulate Inflammation

Recent studies show that 7-hydroxy frullanolide (7HF), an SQ lactone, inhibits CD4^+^ T cell and peritoneal macrophage responses, and lowers CD69 upregulation, IL-2 production and CD4^+^ T cell cyclin upon activation with anti-CD3 and anti-CD28 antibodies [54]. In peritoneal macrophages, 7HF inhibited LPS-induced nitrite and IL-6 production in a Ca^2+^-dependent manner, while in silico studies suggested that 7HF could bind to transient receptor potential cation channel subfamily V member 1 (TRPV1), inositol 1,4,5 thiophosphate receptor (IP3R), and sarco/endoplasmic reticulum Ca^2+^-ATPase (SERCA) channels, which are all important in regulating intracellular calcium fluxes. Binding studies by the same group showed that 7HF inhibited CD4^+^ T cell activation mediated by Ca^2+^-dependent mechanisms, where different Ca^2+^ channels were evaluated after 7HF treatment. Intracellular Ca^2+^ levels were also shown to be regulated by 7HF in CD4^+^ T cells. BAPTA-AM, a specific intracellular Ca^2+^ chelator at suboptimal doses (10 and 100 nM), rescued the 7HF induced decrease of T cell proliferation. When CD4^+^ T cells were treated with Fluo-4 AM followed by 7HF, and activated with anti-CD3 and anti-CD28 antibodies, there was an increase in intracellular Ca^2+^ by 7HF, enhancing CD4^+^ T cell proliferation in vitro. The authors also used a combination of phorbol myristate acetate (PMA) and ionomycin, a system of activation that bypasses the cell surface TCR-CD3 cell signaling pathway. PMA mimics the role of diacyl glycerol, and ionomycin is a Ca^2+^ ionophore. 7HF enhances CD4+ T cell proliferation when cells were only activated with PMA (lower strength of signal), demonstrating its efficacy across different in vitro T cell activation systems. Studies have shown that 7HF inhibited the IL-6 produced by peritoneal macrophages induced by LPS in a Ca^2+^-dependent manner, perhaps by directly binding to TRPV1, IP3R, and SERCA Ca^2+^ channels [166,167,168]. Intraperitoneal administration of 7HF lowered IFNγ and IL-6 in serum, and reduced the effects of dextran sulfate sodium-induced colitis in C57BL/6 mouse models [54]. The results suggest that SQs can function to modulate individual calcium-dependent signaling pathways [166,167]. This could be attributed to the role that SQs play in facilitating cationic permeability of the lipid bilayer and mitochondrial membranes to divalent cations such as calcium that play key roles in several pathophysiological processes [168,169]. Interestingly, calcium has been shown to facilitate the production of SQs in immobilized plant tissues, in response to the extracts of the fungal pathogen *Rhizoctonia solani* [169,170]. The SQ lactone content increases with increasing calcium concentration in nutrient solution, suggesting a positive feedback mechanism.

### 6.2. SQs as Potential Membrane Permeation Enhancers for Drug Delivery Systems

Because SQs promote membrane fluidity, they are potential drug delivery vehicles for therapeutic applications [171]. An interesting observation recently reported by Mishra and colleagues also suggested that the SQ alcohol, cedrol, found in the essential oil of *Juniperus virginiana* (*Cupressaceae*), chemosensitizes human cancer cells and plays a role in suppression of cell proliferation via destabilizing plasma membrane lipid rafts [172]. In addition to plant-derived SQ, there are several studies using SQ derived from marine origin that have demonstrated potential analgesic activity attenuating neuropathic pain, mediating inflammation by the inhibition of TNF, COX-2, and iNOS in RAW264.7 cells, and carrageenan-injected rats [39]. Some other SQs have antineuro-inflammatory and antinociceptive properties in IFNγ-stimulated microglial cells and in neuropathic rats respectively, rendering them as potential therapeutic agents for treatment of neuro-inflammatory diseases.

SQs have anti-inflammatory properties but are lipophilic, cytotoxic, and are not biocompatible with several cell types. Our group has shown that polymeric nanoparticles engineered with poly (lactic-co-glycolic acid) (PLGA) and polyvinyl alcohol (PVA) and encapsulating fukinone, or 10βH-8α,12-epidioxyeremophil7(11)-en8β-ol, an eremophilane-type SQ, delivered controlled SQ cargo to mast cells, the major effector cells of allergic inflammation. The use of natural biodegradable compounds such as PLGA/PVA allowed us to design and control the loading and release of biodegradable SQ materials to cells directly involved in inflammatory reactions. Our studies showed that the use of polymeric nanomaterials improved the biocompatibility of the SQs fukinone and 10βH-8α,12-epidioxyeremophil7(11)-en8β-ol on LAD2 human mast cells. Whereas unencapsulated SQs decreased human mast cell (LAD2) viability by 20%, PLGA/PVA polymeric-encapsulated fukinone, and 10βH-8α,12-epidioxyeremophil7(11)-en8β-ol SQ nanoparticles did not adversely affect LAD2 viability. In addition, SQ encapsulated within nanoparticles successfully delivered the SQ payload to the cells, thus improving the biocompatibility of the SQs. Mast cell response to antigens was modified after treatment with the encapsulated SQ, particularly by decreased gene expression encoding the subunits of the of the high-affinity immunoglobulin E receptor (*FcεR1α*, *FcεR1β*, and *FcεR1γ*), as well as the stem cell factor receptor *(Kit)*. Our studies also showed that encapsulated SQs selectively inhibited tryptase, but had no effect on chymase expression. In addition, PLGA/PVA encapsulated SQ inhibited mast cell degranulation upon IgE/FcεRI stimulation [161].

## 7. Conclusions

Significant research shows that SQ are important regulators of inflammation and that the effects are complex and dependent upon the type, source, and purity of the SQs as well as the tissue and cell type for which the analysis is done. Because SQs have such a complex structure, it is difficult to directly correlate structural changes to the biological effects described in this review. However, some generalizations can be made. SQs can specifically target the NF-kB and NO pathways that lead to inflammation. Furthermore, SQs can directly inhibit the maturation and function of dendritic cells, monocytes, and mast cells, which are important regulatory cells in inflammation, including allergic inflammation. How SQs manifest their effects on these pathways and cells is still not well understood, which is complicated by the varied SQ structures, sources and purities. Studies involving the isolation of SQs from a natural sources uses a unique isolation methods, and often lack rigorous characterization the isolated SQs. As a result, the effects of the SQs or other chemical species isolated in tandem in complex mixtures that are responsible for the biological effects observed is challenging if not impossible to deconvolute. Analyses using highly pure or synthetic SQs are necessary and inevitable.

The correlation of laboratory studies to clinical or therapeutic significance is also challenging. Double-blind and placebo-controlled clinical studies of SQ effects on inflammatory diseases in human cohorts are scarce in the literature (comprehensively examined in several recent reviews, including [173]). Furthermore, even fewer studies exist that identify the appropriate dosage or duration of treatment with SQs, particularly for chronic inflammatory diseases where SQs may be administered over long time periods… SQs have exhibited significant toxic effects on some in vitro models, which raises concerns of adverse events in clinical settings. In light of their diverse chemistry, for SQs to reach therapeutic arenas, rigorous studies at the cellular level in conjunction with appropriate and carefully controlled animal model systems are required in the future.

## Figures and Tables

**Figure 1 molecules-27-02450-f001:**
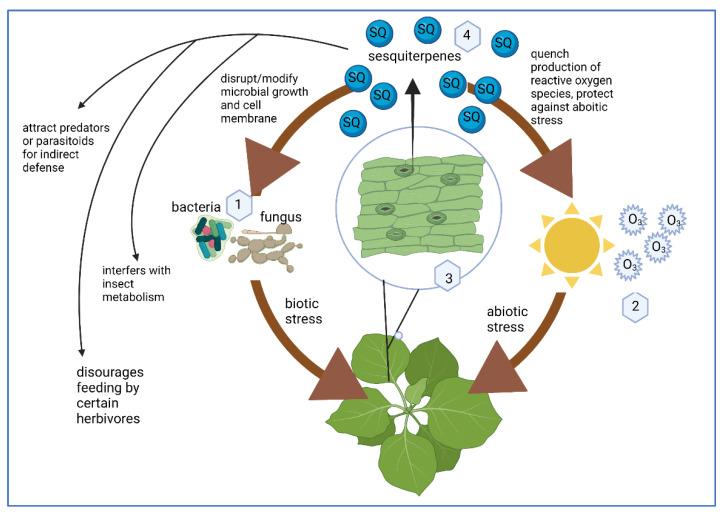
SQ are produced in response to both biotic (1) and abiotic stress (2). (3) SQ are secondary metabolites produced in leaves and flowering heads of the plant but their synthesis amplifies rapidly upon infection (4). Specific mechanisms by which SQ protect plants from both biotic and abiotic stress. Created with Biorender.com.

**Figure 2 molecules-27-02450-f002:**
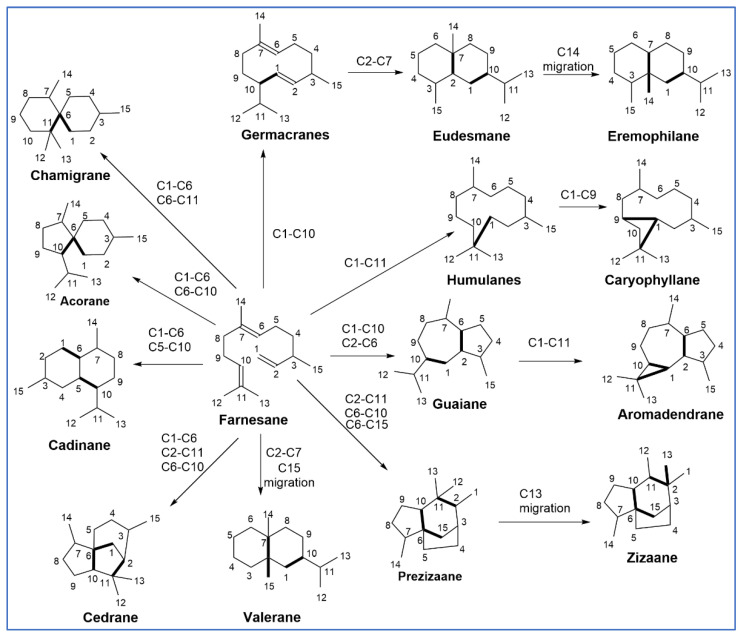
Cyclic SQ family based on chemical rationalization for biogenetic pathway (bold bond shows the new C-C bond formed from farnesane [40].

**Figure 3 molecules-27-02450-f003:**
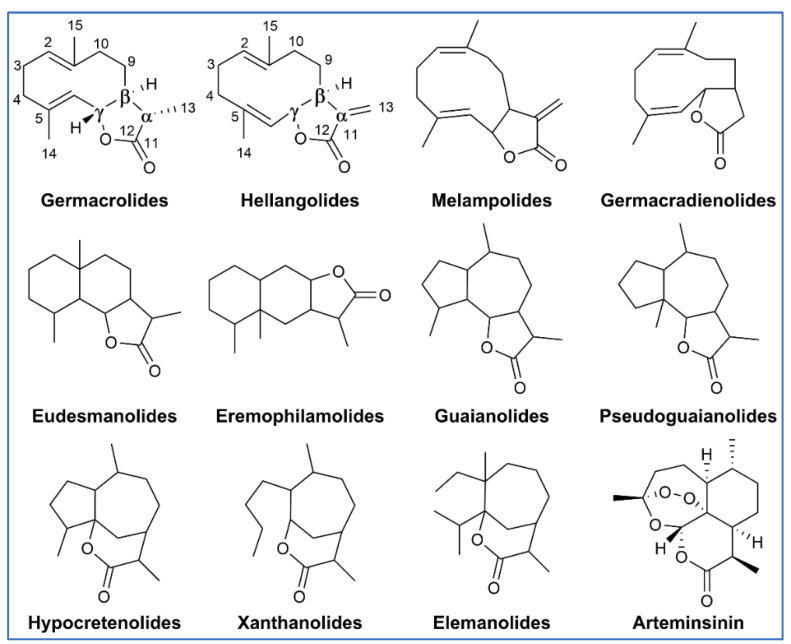
Carbocyclic skeletons of several classes of SQ lactones and chemical structure of Arteminsinin.

**Figure 4 molecules-27-02450-f004:**
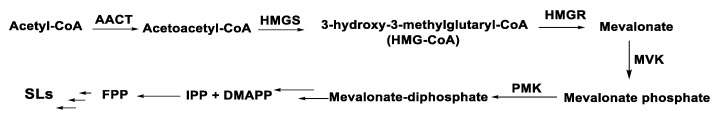
Biosynthesis of SQ lactones via mevalonate pathway [46].

**Figure 5 molecules-27-02450-f005:**
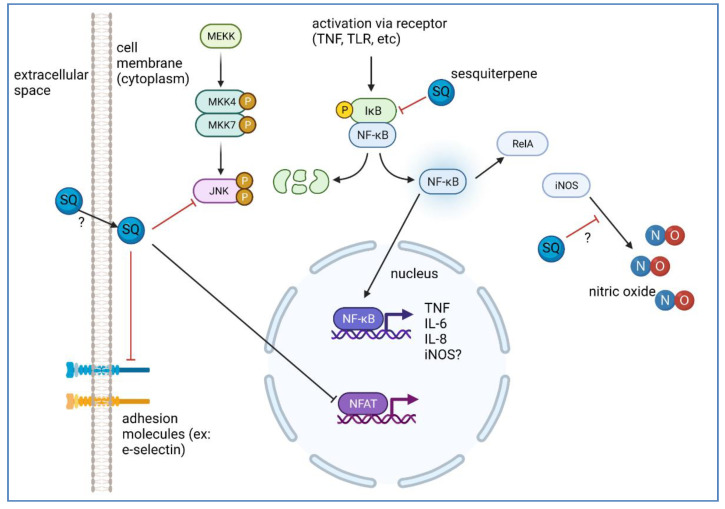
The effect of SQs on inflammation. SQs suppress some inflammatory pathways, inhibiting NF-κB and NFAT activation and blocking expression of endothelial cell adhesion molecules such as e-selectin and ICAM-1. SQs dysregulates IκB phosphorylation, altering the production of pro-inflammatory molecules such as IL-1, IL-6, IL-8, TNF (5), and upstream regulators of the IκB kinase and MEKK, leading to activation of the mitogen-activated protein kinase JNK. SQs inhibit protein expression of iNOS, a regulator of the NF-κB response, and NO production. Created with Biorender.com.

**Figure 6 molecules-27-02450-f006:**
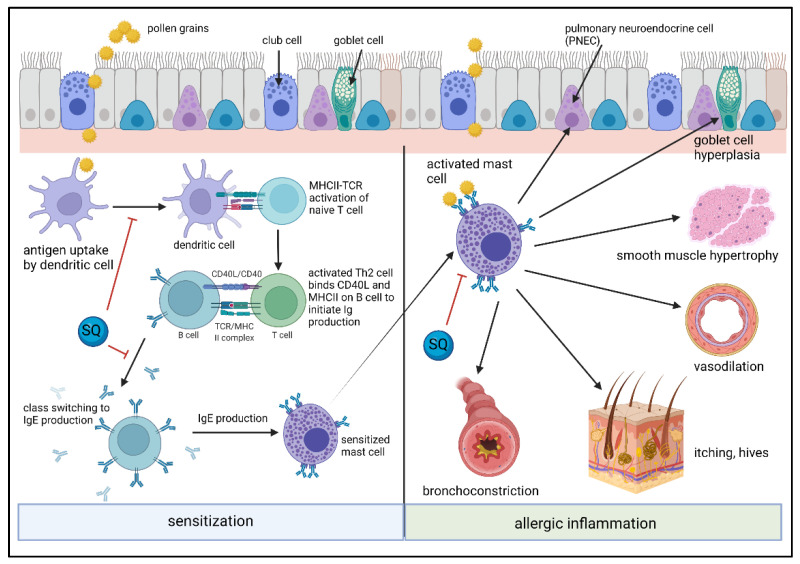
Effect of SQs on dendritic cell maturation, lymphocyte activation, mast cell activation, and allergic inflammation. Allergic sensitization is triggered by allergens such as pollen grains. Once the allergen enters the tissue, either through compromised epithelium or active transport, the antigen is presented by dendritic cells, which then activate naïve T cells to initiate B cells to produce antibodies, including the IgE subclass. The IgE bind to FcεRI receptors on mast cells, activating the mast cells to produce mediators that initiate clinical manifestations of allergic inflammation. Exposure to SQs impairs the differentiation of monocytes into immature dendritic cells. SQs also modify the recruitment of regulatory B cells, attenuating mucosal inflammatory responses. SQs can also modulate allergic inflammation by blocking mast cell activation. Created with Biorender.com.

**Table 1 molecules-27-02450-t001:** Anti-inflammatory effects of sesquiterpenes.

Sesquiterpene (SQ)	Effect	Mechanism
Lob-1, -2, -3, -4, -5, -6, -7, -8 SQ lactones from Neurolaena lobata (74% purity)	Interfere with the induction of inflammatory cell adhesion molecules and chemokines in HUVECtert and THP-1 cells stimulated with bacterial products and cytokines.	Inhibition of LPS and TNF-induced regulation of E-selectin and IL-8 [59]
Vlasouliolides-A, -B, -C, -D, -E, -F, -G, -H,-I SQ lactone dimers from *Vladimiria souliei* (70–80% purity)	Anti-inflammatory activity against LPS-induced NO production in RAW 264.7 cells.	Potent inhibitory activity of the phosphorylation of NF-κB [59,60,61,62].
Neolinulicin-A, and –B SQ dimers from *Inula japonica*	Anti-inflammatory activity against LPS-induced NO production in RAW 264.7 cells.	Inhibition of NO production [63].
8-*0*-methacryloylelephanpane, 2,4-bis-*0*-methyl-8-*0*-methacryloylelephanpane, 4-*0*-ethyl-8-*0*-methacryloylelephanpane, 8-*0*-methacryloylisoelephanpane, 2-*0*-demethyltomenphanatopin C, molephantin A, molephantin B SQ lactones from *Elephantopus mollis* (98% purity)	Anti-inflammatory activity against LPS-induced NO production in RAW 264.7 cells.	Inhibition of pro-inflammatory mediator production such as iNOS, IL-6, MCP-1 and IL-1β through NF-κB and AP-1 signaling pathways [64].
β-elemonic acid from *Boswellia carteri* (94% purity)	Anti-inflammatory activity against LPS-induced NO production in RAW 264.7 cells, mice, and rats.	Inhibition of NO production [65]. Inhibit the overproduction of TNF, IL-6, MCP-1, soluble TNF receptor 1, eotaxin-2, IL-10 and GCSF [65,66]. Inhibition of the activation of NF-κB, by reduced phosphorylation of p65 and attenuates the induction of iNOS, COX-2, NADPH oxidase 2 (NOX-2), and NADPH oxidase 4 (NOX4) leading to the decrease production of NO, PGE2 and ROS. [65]
Dimethylaminoicheliolide (DMAMCL, 82% purity) and Micheliolide guaianolide (MCL, 90% purity)SQ lactones from *Michelia compressa* and *Michelia champaca*	Anti-inflammatory activity against LPS-induced NO production in RAW 264.7 cells. Ameliorates colitis symptoms in a mouse model of dextran sulfate sodium-induced colitis, and azoxymethane/dextran sulfate sodium model of colitis-associated cancer. Anti-inflammatory effects on diabetic kidney disease by inhibiting Mtdh mediated renal inflammation type 2 diabetic *bd/db* mice.	Potent inhibitory activity of the phosphorylation of NF-κB. Significant inhibition of IL-6 and IL-1β, and TNF, and significant decreased of colon tumors [67]. Attenuates inflammatory responses and lipid accumulation in lipid mixture-induced AML12 and LO2 cells by upregulating PPARγ and decreasing phosphorylation of IκBα and NF-κB/p65, inhibiting NF-κB and reducing lipotoxicity [68].
6-0-angeloylplenolin (Brevilin A) from *Centipeda minima* (98% purity)	Inhibition of hepatic stellate cell activation in activated LX-2 cells. Inhibit neuroinflammation in BV2 microglial cells and protects neurones from inflammatory injury. Inhibition of the activation of microglial cells in the hippocampus of mice.	Inhibition of STAT3 phosphorylation through non-SMAD JAK1/STAT3 pathway during the inflammation process following liver injury [52]. Potent inhibitory activity of the phosphorylation of NF-κB and IκB-α [69]. Decreased TNF, IL-1β, and NO, and PGE2 [69]. Inhibition of iNOS, COX-2, NADPH oxidase 2 (NOX-2), and NADPH oxidase 4 (NOX4) leading to the decrease production of NO, PGE2 and ROS [69].
JEUD-38 SQ lactone from *Inula japonica*	Anti-inflammatory activity against LPS-induced NO production in RAW 264.7 cells.	Inhibition of the activation of NF-κB, by reduced phosphorylation of p65 and attenuates the induction of iNOS. Inhibition of the LPS-induced activation of NF-κB, by reduced translocation of p65, via abrogation of IκB-α phosphorylation and degradation. Inhibited LPS-stimulated phosphorylation of MAPKs including ERK1/2, JNK and p38 and attenuates the induction of iNOS [70,71].
Deoxyelephantopin SQ lactone from *Elephantopus scaber* (98% purity)	Glycolysis interference, attenuates LPS-induced IL-1β and high-mobility group box 1 (HMGB1) release in RAW 264.7 cells.	Decreased expression of pyruvate dehydrogenase kinase 1 (PDK1), glucose transporter 1 (GLU1), lactate dehydrogenase A (LDHA), and reduced lactate production. Regulation of the nuclear localization of pyruvate kinase M2 (PKM2) [72].
Ze339 from *Petasites hybridus* (99% purity)	Anti-inflammatory to acute viral infections on primary human nasal epithelial cells (HNECs).	Anti-cytokine effects by interfering with nuclear translocation of STAT-signaling pathways [73].
7-hydroxy frullanolide, SQ lactone from *Sphaeranthus indicus* (98% purity)	Anti-inflammatory activity upon 7HF treated followed by LPS activation of human peripheral blood mononuclear cells.	Downregulates the expression of adhesion molecules such as ICAM1, VCAM1 and E-selectin in TNF activated human endothelial cells. Inhibition of the translocation of NF-κB into the nucleus by inhibiting IKK-β phosphorylation [73,74].
β-caryophyllene bicyclic SQ from *Asparagus falcatus* (98% purity)	Anti-inflammatory effects in rat models of endometriosis. Reduction in cyst size and apoptosis in endometrial explants. Cerebral ischemia-reperfusion injury rat model.	Decreases prostaglandin E2 production, TNF release, nitric oxide synthase and COX-2 [64,69]. Potent agonists for the cannabinoid CB2 receptor [69]. Suppression of IL-1β and TNF [51].
Costunolide (98% purity) and Dehydrocostuslactone (99% purity) SQ lactones from *Laurus nobilis*	Anti-inflammatory.	Inhibition of IL-6 induced tyrosine phosphorylation of STAT3 in human leukemic cell line THP-1. Downregulate phosphorylation of the tyrosine Janus kinases JAK1, JAK2 and Tyk2. Downregulation of NF-κB and STAT3 activation in lung injury mouse model. Counteracts the pro-inflammatory effects of IFN-γ and IL-22 on keratinocytes [75,76,77].
1β-hydroxyalantolactone (IJ-5) SQ lactone from *Inula japonica* (99% purity)	Suppress TNF-induced NF-κB activation and inflammatory gene transcription. Attenuate atopic dermatitis severity, IgE, IL-4, and IL-6 in serum, mRNA levels of TNF, IL-1, IL-4 and IL-6 in skin lesions in vivo.	Inhibition of the ubiquitination of receptor-interacting protein 1 and NF-κB essential modifier [78]. Inhibition of inflammatory cytokine expression through NF-κB activating pathway [79].
Alantolactone (AL, 95% purity) and isoalantolactone (IAL, 95% purity) SQ lactones from *Inula helenium*	Inhibition of TNF-induced activation of synovial fibroblasts, and RAW 264.7 cells.	Inhibition of TNF-induced activation of NF-κB and MAPK pathways, supress the expression of MMP-3, MCP-1, and IL-1, IL-6 and iNOS [79].
Fukinone and 10βH-8α,12-epidioxyeremophil-7(11)-en-8β-ol isolated from *Petasites tatewakianus*	Inhibit dendritic cell maturation and activation.	Dendritic cell inhibition is mediated by nuclear peroxisome-activated receptor γ (PPARγ) [80].

**Table 2 molecules-27-02450-t002:** Sesquiterpene effects on mast cells.

Sesquiterpene (SQ)	Effect	Mechanism
Fluvastatin	Inhibited RBL-2H3 cells degranulation.	Ca^2+^ independent due to suppression of geranylgeranyl transferase [141].
Parthenolide	Inhibited RBL-2H3, and BMMC degranulation. Inhibited passive cutaneous anaphylaxis reaction in mice.	Disrupted microtubule formation-fyn kinase dependent [142]
Magnolialide	Inhibited RBL-2H3 cells degranulation.	Decreased levels of IL-4 [136]
Bakkenolide B	Inhibited RBL-2H3 cells degranulation.	Suppressed IL-4 release [143]. Inhibited NOS2 and COX-2 [143]. Suppressed IL-4 production [136,144].
(-)-Elema-1,3,11(13)-trien-12-ol	Inhibit RBL-2H3 cells degranulation.	Suppressed IL-4 production [144].
Thujopsene	Inhibit RBL-2H3 cells degranulation.	Suppressed IL-4 production [144].
Atractylenolide III	Inhibit RBL-2H3 cells degranulation.	Inhibit phosphorylation of Lyn, Fyn, Syk, LAT, PLCγ, Gab2, Akt, p38, and JNK kinases; Ca^2+^ dependent [145].
Artesunate	Reduce infiltration of mast cell in mouse skin. Inhibit the release of inflammatory cytokines, downregulate Th17 cell responses in RBL-2H3 and mature human cultured mast cells.	Inhibited IgE-induced Syk and PLCγ1 phosphorylation, production of IP3, and rise in cytosolic Ca^2+^ level [146]. Reduce IgE and TNF concentration in serum. Suppress of IL-6, IL-17, and IL-23 expression [147]. Promote SOCS3 protein and inhibit ROR-γt protein and STAT3 phosphorylation [96].
Aegeline	Inhibit degranulation and cytokine secretion of RBL-2H3 cells.	Influence the intracellular Ca^2+^ pool [148].
Artekeiskeanol A	Inhibit degranulation and cytokine secretion of RBL-2H3 cells.	Suppress TNF, IL-13 and phosphorylation of Akt, p38 MAPK, JNK, p44/42MAPK [149].
Tussilagone	Inhibit degranulation and cytokine secretion of RBL-2H3 cells.	Suppress phosphorylation of Lyn, Syk, Akt, NF-κB p65, ERK and p38 MAPK [150].
SQ lactones derived from 6β-angeloyloxy3β,8-dihydroxyeremophil-7(11)-en-12,8β-olide (F-1)	Inhibit degranulation and cytokine secretion of RBL-2H3 cells.	Inhibit β-hexosaminidase release and TNF production [140].
3-butyl-1-chloro-8-(2-methoxycarbonyl)phenyl-5H-imidazo[1,5-b]isoquinolin-10-one (U63A05)	Inhibit degranulation and cytokine secretion of RBL-2H3 and BMMC.	Inhibit Syk activation; Ca^2+^ independent [151,152].
Cacalolides	Inhibit degranulation and cytokine secretion of BMMC.	Inhibit the activity mediated by FcεRI-induced intracellular Ca^2+^ mobilization, ROS production, VEGFR-2, and activation of PI3K-Akt kinases, and MAPK pathway [153].
Atractylone	Decrease histamine levels, IgE, IL-4, IL-5, IL-6, VEGF, and IL-13 in peritoneal mast cells of PCA-induced mice Attenuate pro-inflammatory cytokine production and mRNA expression of phorbol 12-myristate 13-acetate and calcium ionophore A23187-stimulated HMC-1, rat peritoneal mast cells, and allergic rhinitis mouse model.	Inhibit intracellular Ca^2+^, tryptase release, and histamine release. Decrease histidine decarboxylase activity and expression. Induce caspase-1/NF-κB/MAPKs activation. Reduce total IgE, histamine, PGD2, TSLP, IL-1β, IL-4, IL-5, IL-6, IL-13, TNF, COX-2, ICAM-1, and MIP-2 [153,154].
Britanin	Inhibit pro-inflammatory cytokines and degranulation of HMC-1 and BMMC.	Suppress gene expression and secretion of pro-inflammatory cytokines [155]. Attenuate activation of NF-κB pathway. Inhibit generation of PGD2 and phosphorylation of Syk-dependent pathway [155,156].
β-Eudesmol	Inhibit the production and expression of IL-6 in PMA and Ca^2+^ ionophore-stimulated HMC-1; suppress SCF-induced mast cell migration and morphological.alterations, reduce F-actin formation in rat peritoneal mast cells.	Suppress activation of p38 MAPKs, and NF-κB. Suppress the activation of caspase-1 and expression of receptor-interacting protein-2. Reduce activation of Fyn kinase, Rac1 GTPase, and p38 MAPKs [154,157] {Nam, 2017 #773}.
Dehydroleucodine xanthatin	Inhibit degranulation of LAD2, rat peritoneal mast cells and rat gastric mucosa mast cells.	Anti-inflammatory properties, with inhibition of mast cell activation [158,159,160].
Fukinone	Inhibit IgE dependent degranulation.	Inhibit expression of FcεRI(α, β,γ), and *Kit* receptors, and tryptase expression [161].

## Data Availability

Not applicable.

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
