# Peer review of "The Complexity of Sesquiterpene Chemistry Dictates Its Pleiotropic Biologic Effects on Inflammation"

_molecules, 2022, doi:10.3390/molecules27082450_

Round 1
Reviewer 1 Report
Arizmendi et al. have presented a review article that emphasizes the chemical as well as inflammatory properties of sesquiterpene. Overall the paper is well written however in some sections, there are serious concerns which decrease the quality of this review. I have a concern regarding the novelty of this concept as there are several articles which explains the same concept and also, author have incorporated some date which is not required. In my opinion, the manuscript cannot be published in this journal without resolving some major issues regarding the quality and data of this review.
- Please justify the novelty of this concept. In which manner this review article is different from other.
- Authors are suggested to reformat the content of table. Due to overlapping of data, tables are hard to understand.
- Line number 12, “Sesquiterpenes (SQ) are volatile compounds made by plants, insects and marine organisms”. In this sentence authors have shown that SQs are produced by all these but they described only plant derived SQ. Any specific reason?
- Line number 12-13, SQ have a large range of biological …………………………….kappa-B (NF-B) signaling path, please reformat.
- In introduction section, there are several places where author discussed about previous literature but not even a single reference was found or cited in the text.
- Again in section 2, authors have mentioned that SQs are produced in nature and in nature all the three sources should be considered otherwise change the title of this section.
- If authors have focused their study only on chemical and inflammatory properties then there is no need of section 2.2
- In table 1 and 2, column representing the effect of some compounds is blank, please explain.
- In section 4, authors have explained the mechanism of SQs, please justify the correlation of section 4 with section 4.1 “Isolation and purification of SQ”
- Authors are suggested to add a table for describing the abbreviations used throughout the manuscript.
Reviewer 2 Report
The manuscript, “The complexity of sesquiterpene chemistry dictates its pleiotropic biologic effects on inflammation”, reports the effect of Sesquiterpenes on inflammation in the context of their complex chemistry. In my opinion, the manuscript is suitable for publication in Molecules, after the authors have addressed the following comments and questions:
- Line 14: “…Since SQ can be isolated from over 1600 genera and 2500 species grown worldwide, they are an attractive source of phytochemical therapeutics”….and line 29 “…SQ can be isolated from Asteraceae and Compositae plant families from over 1600 genera and 2500 species, using a variety of approaches including silica gel filtration and chromatography.”…. So, do the 1600 genera and 2500 species come only from Asteraceae and Compositae? Please, make this information clearer
- Could you identify the family name for each species in the paper? Inula helenium (Asteraceae).
- Figure 2 should be improved.
- Line 210 “…… Second, SQ can be poorly bioavailable at low concentrations but cytotoxic at high concentrations”….. Could you elaborate on that?
- Line 502. Farfarae flos is the species? Could you confirm that the species is Tussilago farfara? If so, could you add this information? Suggestion: "Another SQ compound, tussilagone isolated from Farfarae flos (Tussilago farfara, Asteraceae)”
- Figure 6 should be improved. Format the font size. The overlap of names and drawings should be avoided.
- Line 575. “…A recent study demonstrates a role of marine sponge-derived SQ…” It would be interesting for the authors to demonstrate in the introduction the keywords used to make the paper. The reading of the paper was ambiguous, in the abstract, I understood that it was a general review. In the introduction, are the SQs described in this paper only from Asteraceae and Compositae? And, in line 575 it appears SQ of marine sponge. Could you elaborate on that?
Round 2
Reviewer 1 Report
Authors have successfully incorporated all the suggestions provided therefore this manuscript can now be accepted in present form.